# The Investigation on the Dark Sector at the PADME Experiment †

**Fabio Ferrarotto** ‡ 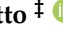

INFN Sez. Roma 1, University "La Sapienza", P.le A. Moro 2, 00185 Rome, Italy; fabio.ferrarotto@roma1.infn.it; Tel.: +39-06-4991-4417

† This paper is based on the talk at the 7th International Conference on New Frontiers in Physics (ICNFP 2018), Crete, Greece, 4–12 July 2018.

‡ On behalf of the PADME Collaboration.

**Abstract:** In this paper, we present the design and expected performance of the various detectors of the PADME experiment. The experiment design has been optimized for the detection of the final state photons produced along with a "Dark Photon", decaying to invisible particles, in the annihilation a of 550 MeV positron with an atomic electron of a thin target. The PADME experiment has been built in a new dedicated experimental hall at the Beam Test Facility (BTF) of the INFN Frascati National Laboratories and has been taking data since the third quarter of 2018.

**Keywords:** dark matter detectors (WIMPs, axions, etc.); calorimeters; Cherenkov detectors

---

## 1. A Short Introduction to Dark Matter and the "Dark Photon"

From cosmological and astrophysical observations of gravitational effects, something else than ordinary baryonic matter should exist. The abundance of this new entity is estimated to be five times larger than Standard Model (SM) particles in terms of mass.

Dark Matter (DM) particles may also manifest through interactions other that gravity; in this case, it could be produced in SM interactions. Many extensions of the SM provide DM candidates with the features of Weakly Interacting Massive Particles (WIMP) that accommodate the gravitational evidences and might be in the reach of current experiments. There are long standing experimental programs searching for interactions of cosmic DM with high-mass and low-background experiments and an intense scrutiny of data at high energy accelerators looking for signs of direct production of DM candidates. However, thus far, none of these investigations provide convincing evidence for a DM candidate [1,2].

An alternative class of simple models adds an additional $U(1)$ "hidden" symmetry to the SM: $U(1)_Y + SU(2)_{Weak} + SU(3)_{Strong} + U(1)_{A'}$, and assumes that the corresponding gauge boson has a light mass. This new particle ($A'$, typically referred to as "Dark Photon", DP) can act as a portal connecting the hidden sector to the SM particles because of its possible mixing with the SM photon [3,4]. The small mass is still compatible with limits from direct detection and an $A'$ boson might explain anomalies in muon magnetic moment, results from scattering experiments, searches for DM and antimatter excess in cosmic rays.

The effective interaction between SM fermions and the "Dark Photon" is parameterized in terms of a factor $\epsilon$ representing the mixing strength and is QED-like.

Not all SM particles need to be charged under this new symmetry. In the most general case, the new effective charge $q'_f$ may be different between leptons and quarks and can even be 0 for quarks [5].

If the coupling of the SM particles to $A'$ arises only via the kinetic mixing, the coupling is universal and proportional to the electromagnetic charge $q_{em}$. In this case, a single additional parameter $\epsilon$ is needed to describe the $A'$ phenomenology in addition to its mass.

The $A'$ may be produced by processes similar to the standard QED annihilation, bremsstrahlung and meson decay. Two scenarios are possible for the "Dark Photon" decay:

- If no DM particles exist with $m_{DM} < m_{A'}/2$, the $A'$ decays to SM particles with a BR depending on $m_{A'}$; the $A'$ lifetime is proportional to $1/(\alpha \epsilon^2 m_{A'})$.
- If DM particles exist ($\chi$) with $m_{DM} < m_{A'}/2$, the $A'$ decays to invisible DM particles with likely BR $\sim 1$. The SM decays are suppressed by a factor $\epsilon^2$ and the $A'$ lifetime is $\approx 1/(\alpha_D m_{A'})$ where $\alpha_D$ is the $A'$ coupling constant to the Dark Sector.

The PADME experiment will search for a light $A'$ using a positron beam on a thin active target, detecting the photons produced in the annihilation reaction $e^+e^- \to \gamma A'$ and measuring the event missing mass for the invisible DP decay.

After the first proposal [6], the PADME experiment at the DAΦNE Linac [7] Beam Test Facility (BTF) of the Laboratori Nazionali di Frascati (LNF) was approved by INFN in 2015 with the goal of searching for this new mediator $A'$.

The collaboration is composed by the following institutions: INFN Roma and "La Sapienza" University of Roma, INFN Frascati, INFN Lecce and University of Salento, MTA Atomki Debrecen, University of Sofia, Cornell University, US William and Mary College and Iowa University.

## 2. PADME Physics Cases

Experiments have shown a discrepancy of about 3 $\sigma$ between theory and experiment for the anomalous muon magnetic dipole moment $g - 2$, a very well calculated quantity. Contributions may come from additional graphs containing a "Dark Photon" [8].

This was one of the strongest motivations for the DM models with a hidden sector and a "Dark Photon" and a strong starting motivation for the PADME experiment, although not the only one. As is visible in Figure 1a, in 2017, Babar [9] put a stringent limit on the dark photon production phase space, excluding the simplest models for the $(g - 2)_\mu$ contribution.

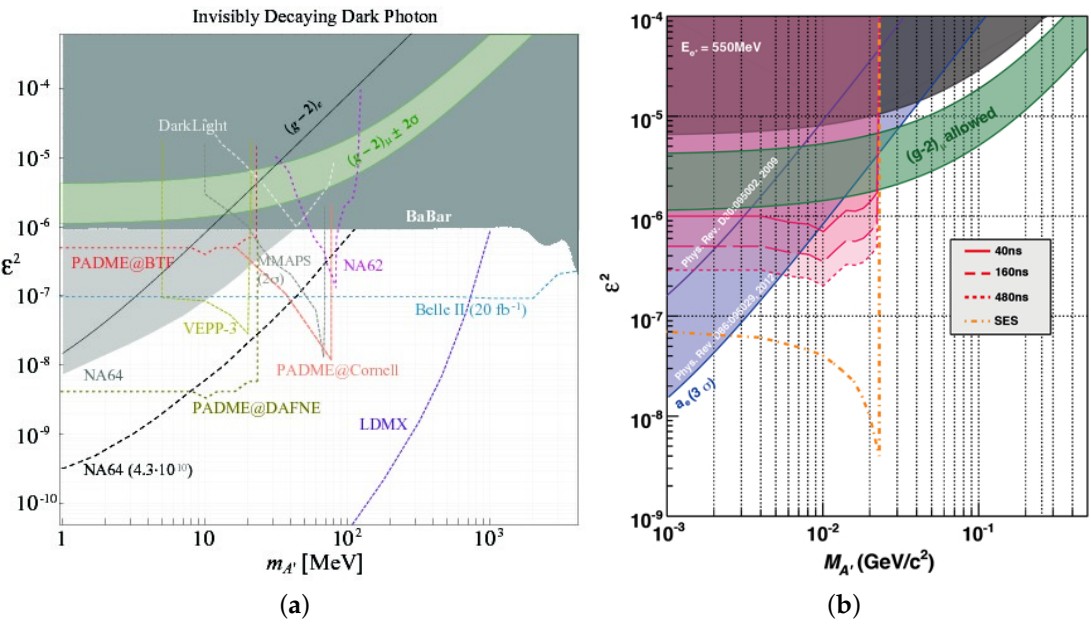

**Figure 1.** (**a**) Actual limits for invisible decay of DP; and (**b**) Padme sensitivity limits for $10^{13}$ POTs and different bunch lengths.

PADME can search also for long lived Axion-Like Particles (ALPs: *a*) [10] produced through a virtual off-shell photon. In the mass region <100 MeV, ALPs are long lived and would manifest in invisible decay via missing mass, thus falling in the same search for the "Dark Photon". In the visible decay modes $a \rightarrow \gamma\gamma$ and other production mechanisms could be explored. The observable final states at PADME will be $e\gamma\gamma$ or $\gamma\gamma\gamma$. Even without any selection cut, PADME will be background free for ALP masses > 50 MeV.

An experiment on excitation of $^8Be$ by protons on $^7Li$ target and $\gamma$ decay [11] has observed a 6.8 $\sigma$ excess in symmetric $e^+e^-$ pairs, which can be interpreted as a new boson with mass ∼16.6 MeV, with evidence for resonant production. This new particle, according to Feng et al. [12,13], is not compatible with present limits unless it is a proto-phobic vector boson or a boson with axial couplings to quarks.

The Na64 experiment has recently put a limit [14] on the production of this new particle, but still allows a fair window in the production phase space where PADME can look for its production.

The production of such boson can be tested by PADME at LNF with production at threshold by tuning the $e^+$ beam with variable energies at ∼283 MeV to produce it near threshold [15], with the further advantage, with respect to a "beam-dump"-like experiment, of being able to "tag" the associated decaying charged particles to measure the meson invariant mass.

## 3. The PADME Approach

The PADME strategy for the search of an $A'$ consists in firing a positron beam of 550 MeV energy ($M_{A'} < 23.7$ MeV) with ∼20,000 particles in a bunch length of 200 ns at 49 Hz repetition rate, provided by the LNF's LINAC, on a thin active diamond target [16]. This will be done by detecting the SM photons produced in the annihilation reaction $e^+e^- \rightarrow \gamma A'$ and measuring the event missing mass $M^2_{miss} = (p_{e^+} + p_{e^-} - p_\gamma)^2$ for the invisible decay.

Figure 2 shows a 3D view of the PADME experiment with all detectors outlined, with a "Golden Signal" event: one single $\gamma$ observed in the main Electromagnetic Calorimeter (ECAL) and nothing in all other detectors in a narrow time window ($\pm$ 2 ns).

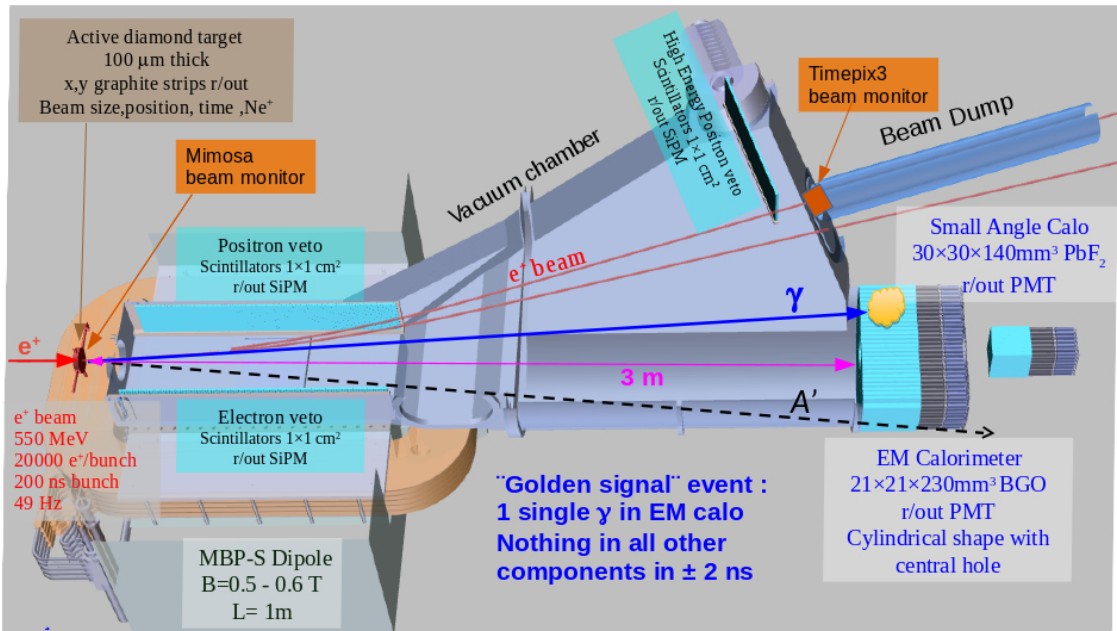

**Figure 2.** Padme detector 3D layout with all basic components explained. In the figurem we show also a "Golden" event in the detector.

The positron momentum is determined by the accelerator characteristics and beam resolution, thus the missing mass resolution is determined by the annihilation point, $E_\gamma$ and $\Theta_\gamma$. The only assumption we make in the experiment is that $A'$ couples to the electron.

We exploit a clear two-body correlation, thus minimizing the backgrounds and obtaining the best possible resolution on energy/angle measurement. The dominant process in $e^+e^-$ interactions with matter is bremsstrahlung, so the photon and positron vetoing is also crucial.

As shown in Figure 2, to detect with high efficiency and precision the photon produced in coincidence with an $A'$ particle, and to reject all the others due to background events, there are two calorimeter units in the experimental setup: the main one is the Electromagnetic Calorimeter (ECAL), while the second one is the faster Small Angle Calorimeter (SAC).

The active target is housed in a cross at the end of the beam line, after which the beam enters a large vacuum chamber, which houses the charged particle detectors (P/E vetoes, to detect background and decaying particles), situated inside a large dipole magnet from the CERN SPS transport line (1 m length, 23 cm vertical gap, 0.55 T magnetic field).

At 3 m distance from the target, before the ECAL, we have a large (630 mm diameter) carbon fiber window, which is the exit from the vacuum chamber, to minimize beam interactions in front of the calorimeters.

Beam monitor detectors, to precisely measure the beam characteristics at run time, have been inserted at the entrance (MIMOSA) and at the exit (TimePix3) of the vacuum chamber; the latter is located behind a thin Al window flange, to minimize interactions, which is the position for the beam-dump. To measure the tails of the exit positron beam-dump, the Timepix3 is flanked by the HEP-veto detector starting just at one of its end corner.

### 3.1. Backgrounds

The main backgrounds in the experiment are:

- bremsstrahlung in the field of the target nuclei, which dominates the region of the high squared missing mass: photons mostly have low energies and small angles;
- two photon annihilation which peaks at $M_{miss}^2 = 0$ and the gammas are quasi-symmetric in angle for $E_\gamma > 50$ MeV;
- three photon annihilation: the symmetry in the two-photon system is lost with a decrease in vetoing capabilities, as these events do not peak in $M_{miss}^2$;
- pile-up: important contribution, but rejected by the cuts on maximum cluster energy and $M_{miss}^2$; and
- radiative Bhabha scattering: topology close to bremsstrahlung.

The main signal selection cuts are:

1. only one cluster in the ECAL fiducial volume;
2. no hits in the E/P/HEP vetoes in $\pm$ 2 ns;
3. no $\gamma$ in the SAC with $E_\gamma > 50$ MeV in $\pm 2$ ns; and
4. cut $20 \div 150 MeV < E_\gamma < 120 \div 350$ MeV (depending on $m_{A'}$).

The E/P/HEP Vetoes and the SAC are essential to veto all backgrounds.

### 3.2. PADME Sensitivity

Limits from actual and future experiments for the search of an invisible "Dark Photon" are shown in Figure 1a, including limits from PADME at BTF with 200 ns bunch length.

Using GEANT4 [17], we simulated $2.5 \times 10^{10}$ events with 550 MeV $e^+$ on the thin carbon active target [18]. The number of background events was then extrapolated to $1 \times 10^{13}$ positrons on target (POT). Limits were obtained assuming six months of data taking at 70% efficiency with a bunch length of 200 ns at 49 Hz = $1 \times 10^{13}$ POT (20,000 $e^+$/bunch $\times 0.5 \times 3.1 \times 10^7$ s $\times 0.7 \times 49$ Hz).

In Figure 1b, we show the expected limits for PADME with different bunch lengths: PADME can explore in a model independent way the region down to $\epsilon \approx 10^{-3}$ for $m_{A'} < 23.7$ MeV ($E_{beam}$ = 550 MeV—LNF Linac).

With only minimal assumptions, the coupling of any new light particle produced in $e^+e^-$ annihilation may be measured or constrained: "Dark Photon", Axion-Like Particles, Dark Higgs, new proto-phobic vector boson, etc.

## 4. The PADME Detectors

### 4.1. Diamond Active Target

Diamond is the solid material with the best $ee(\gamma\gamma)$/Bremsstrahlung ratio (Z = 6). The active target is made of a large CVD polycrystalline diamond, produced by Applied Diamond Inc. (Wilmington, DE, USA), $20 \times 20$ mm$^2$ wide and 100 μm thick, with $16 \times 1$ mm graphitized strips on each side and X-Y readout, all in a single detector.

The active target is used to monitor at each bunch the beam spot position and the number of particles hitting the target. The signals from each strip are readout by low-noise Charge Sensitive Amplifiers (CSA) integrated in the 16 channel chip AMADEUS from IDEAS. Final commissioning and calibration have been successfully done.

For a prototype studied in a beam test at BTF in 2015 [16] using 5000 $e^+$/bunch, we measured:

- a charge collection distance of about 11 μm;
- a beam centroid spatial resolution of ~0.2 mm; and
- a timing resolution of ~0.7 ns per strip.

At the start of the physics run, the diamond active target will be brought from the parking position to the nominal position on the beam line, after taking off the beam monitor detector (MIMOSA), which provided preliminary diagnostics of the beam.

### 4.2. Electromagnetic Calorimeter

The Electromagnetic Calorimeter (ECAL) is the main detector of PADME. Its final design is a compromise between performances, dimensions and cost: the ECAL material choice is crucial for the experiment performances.

The final choice was to use BGO crystals, obtained machining the crystals recovered from one of the end-caps of the electromagnetic calorimeter of the dismantled L3 experiment at LEP [19].

BGO features high light yield, high density, small radiation length $X_0$, small Moliére radius, long $\tau_{decay}$.

The ECAL (see Figure 3a) is composed of 616 crystals each $21 \times 21 \times 230$ mm$^3$ for a depth corresponding to 20.5 $X_0$ with an approximate cylindrical shape with a radius of ≈285 mm. The Moliére radius of BGO is 22.6 mm, thus compatible with the crystal size.

It has a central square inner hole $105 \times 105$ mm$^2$ (five units wide) meant to let the Bremsstrahlung radiation (which is sharply peaked at small angles) to pass through and to be vetoed by the faster SAC calorimeter placed behind; this is necessary since the BGO has a long scintillating light decay time ($\tau \sim 300$ ns) and would be continuously blinded by this radiation.

The crystal readout is made by HZC XP1912 PhotoMultipliers (PMT) [20] with 19 mm diameter, ensuring a good coverage; PMTs have been glued to the crystals and painted white diffusive (see Figure 3b). Fifty-micrometer Tedlar foils have been put between layers to avoid optical crosstalk between crystals.

The High Voltage (HV) system is based on a CAEN SY4527 unit housing A7030N modules (48 channels, negative 3 KV, $I_{max}$ = 1 mA) distributed via dedicated supply boxes to each PMT.

The PMT readout is made by CAEN V1742 digitizers [21] at 1 GS/s (1024 samples ~1 μs), recording almost the entire BGO crystal signal detected during the bunch length (see Figure 4a). Before assembling the ECAL, all crystals have been tested and HV calibrated using a $^{22}$Na source.

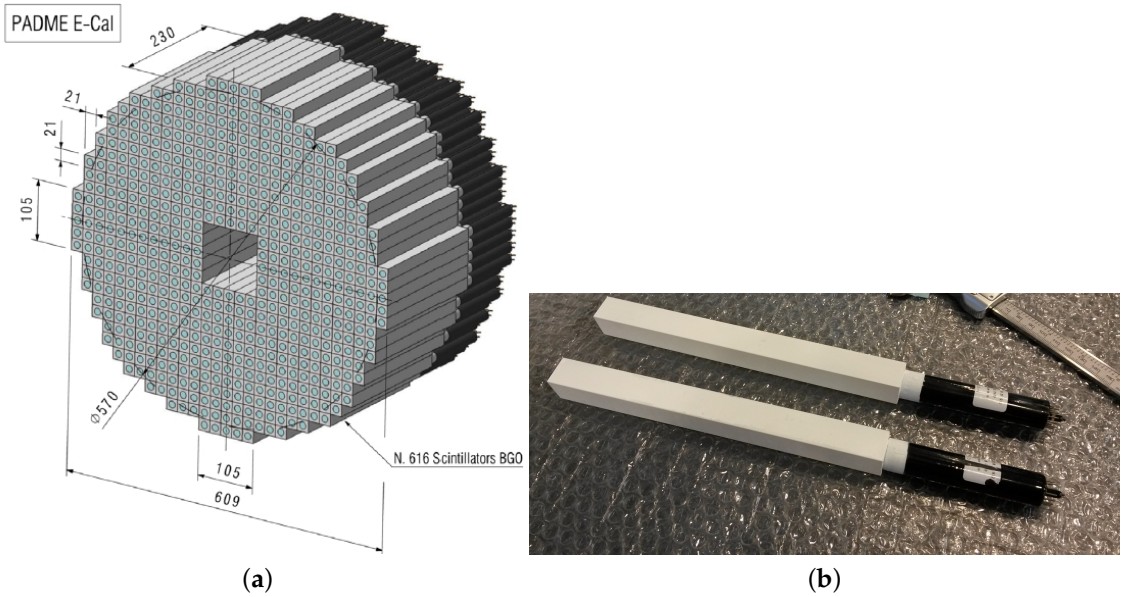

(**a**)                              (**b**)

**Figure 3.** (**a**) ECAL design with dimensions in mm; (**b**) HZC XP1912 PMTs painted and glued to BGO crystals.

Performance of a 5 × 5 crystals prototype was studied in dedicated beam tests at BTF in 2016 [22] and 2017 and showed:

- energy resolution $\sigma(E)/E = 2\%/\sqrt{E} \oplus 0.003\%/E \oplus 1.15\%$ where E is in GeV (see Figure 4b);
- angular resolution $\sigma(\Theta) < 1$ mrad;
- timing resolution < 1 ns from signal shape fit;
- good linearity up to ~1 GeV;
- charge collection of ~16 pC/MeV; and
- good reproducibility of the test results with our MonteCarlo.

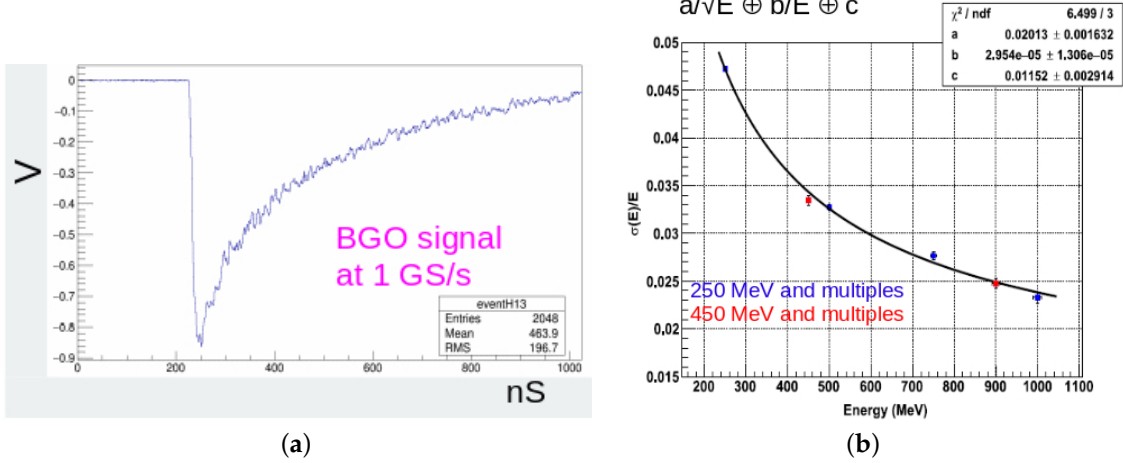

(**a**)                              (**b**)

**Figure 4.** (**a**) BGO signal digitized at 1 GS/s; and (**b**) ECAL 2016 BTF test energy resolution [22].

These results are in perfect agreement with expectations for our calorimeter design and are adequate for the experiment goals.

In Figure 5b, a side view of the ECAL as mounted in the experimental hall together with the SAC with its final support structure is shown.

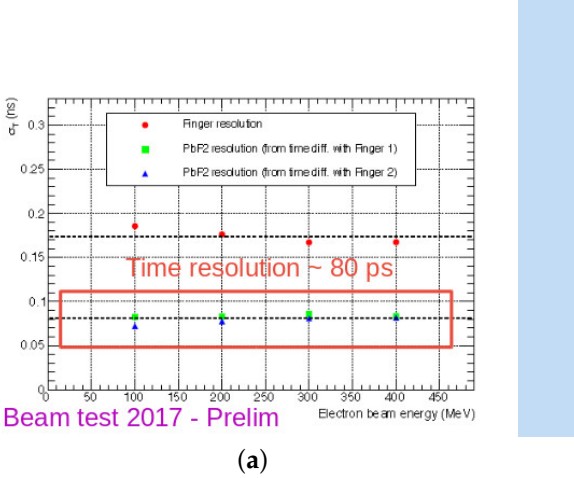
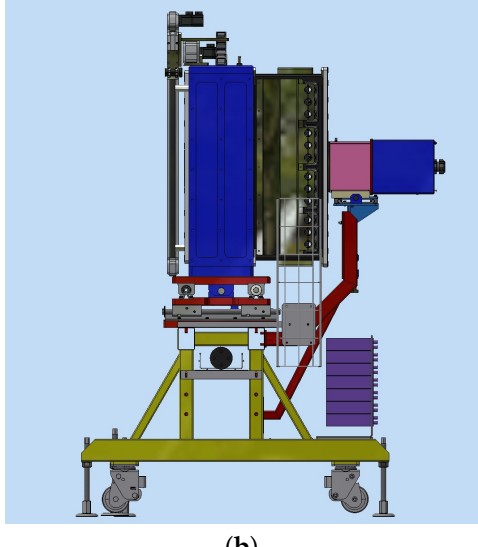

<div align="center">(<b>a</b>)                               (<b>b</b>)</div>

**Figure 5.** (**a**) SAC BTF test time resolution [23]; and (**b**) final SAC mounting (pink area on the right—followed by blue area of PMTs) together with the ECAL (blue area at center—followed by grey area of PMTs box) with all mechanical structure, including the HV distribution boxes in the right lower part (violet area) and the mechanical support structure (light green).

### 4.3. Small Angle Calorimeter

The basic requirements for this calorimeter/veto detector are:

- be fast and compact;
- measure $E_\gamma$ from ∼50 MeV and have a double-peak separation resolution capable of distinguishing several dozen photons in a 200 ns time span;
- avoid scintillation mechanism ($\tau$ too long), thus use a Cherenkov light detector to get the needed time resolution O ($\leq 200$) ps, which needs very fast photosensors;
- be transparent at shorter wavelengths (higher Cherenkov yield);
- be radiation tolerant: O (1 Gy) per $10^{13}$ $e^+$ on target;
- no need for high light yield material: 0.5–2 p.e./MeV is OK; and
- have a moderate energy resolution O (5–10 %)/$\sqrt{E}$.

The final choice was a $5 \times 5$ matrix of $30 \times 30 \times 140$ mm$^3$ PbF$_2$ crystals, slightly larger than the central square hole of the ECAL. Being based on Cherenkov counters, the SAC has a short dead time (∼3 ns) and a rate capability 10 times better than ECAL. A charge collection of ∼1–2 p.e./MeV is expected, with black wrapping of crystals to avoid reflections inside.

The readout is done using Hamamatsu R13478UV PMT (BA): fast (< 1 ns rise time) with a diameter of 2.54 mm (56% surface coverage). The SAC and ECAL share the same high voltage and readout system, where signal sampling is used at 2.5 GS/s.

To determine the time resolution of this detector, time-of-flight measurements were performed in a beam test at BTF in 2017 [23] using a scintillator finger with a measured time resolution of 174 ps as start counter. The measurement was affected by a resolution interpreted as the sum in quadrature of the intrinsic resolution of the SAC crystals, which turned out to be ∼81 ps (see Figure 5a), and the resolution of the finger. The SAC energy resolution was measured to be $\sigma(E)/E \sim \sqrt{p_1^2/E + p_0^2}$ ($p_1$ = 5.7%, $p_0$ = 6%). We also measured a light yield of 2.07 photo-electrons per MeV.

These performances match very well with our requirements for this calorimeter.

Figure 5b shows the SAC as is mounted in the experimental hall in the final position behind the ECAL with the full mechanical support structure.

### 4.4. Charged Particle Detectors

To detect and veto charged particles inside the large vacuum chamber and the magnet (low energy $e^+/e^-$), we foresee two dedicated detectors (Pveto and Eveto) and, close to the beam dump exit (high energy $e^+$), another detector (HEPveto). These are all made by plastic scintillator bars $11 \times 10 \times 178$ mm$^3$ with a 1.2 mm diameter Wave Length Shifter (WLS) fiber placed in a grove along the slab. The three detectors are made, respectively, by: 96 scintillator bars (Eveto, Pveto) and 16 bars (HEPveto).

The readout is performed by $3 \times 3$ mm$^2$ Hamamatsu S13360 Silicon PhotoMultipliers that collect the light from both scintillator and fiber. The position of the hit gives a rough estimate at the percent level of the particle momentum and we measure also its timing with the same readout electronics of ECAL and SAC (at a speed of 2.5 GS/s).

A big challenge in the design has been that the detectors are all inside the vacuum chamber and the first two are inside the magnetic field region.

In dedicated beam tests at BTF in 2017 [24], we met the main requirements:

- time resolution ∼300–500 ps; and
- efficiency better than 99.5% for MIPs.

In Figure 6a, the Eveto and Pveto are shown mounted on their mechanical structure inside the vacuum chamber, complete with front-end electronics and cabling.

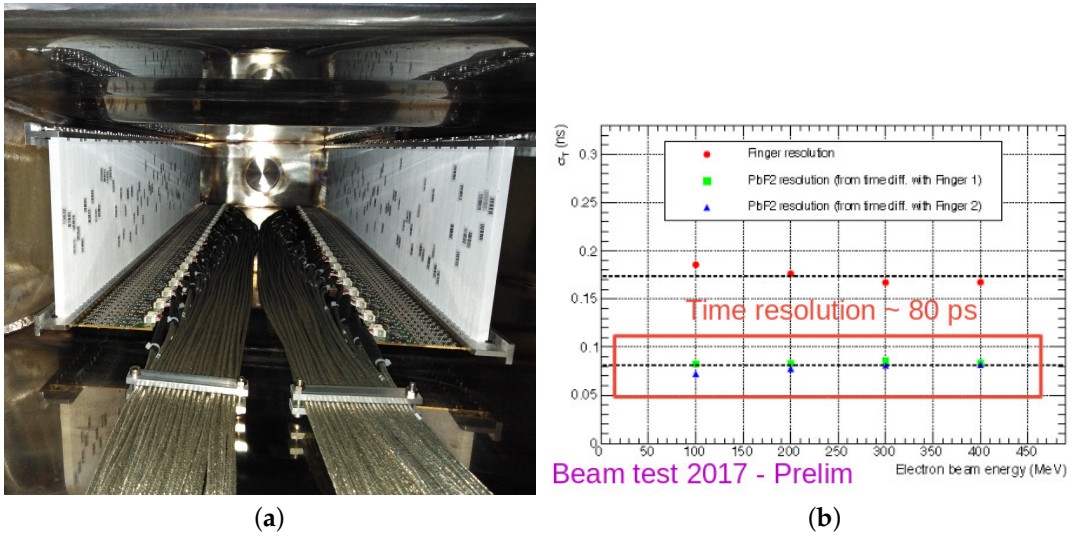

(**a**)                                                                                  (**b**)

**Figure 6.** (**a**) Mounted Eveto and Pveto inside vacuum chamber; and (**b**) MIMOSA setup with full cross.

### 4.5. Beam Monitors

PADME needs to measure both beam divergence and beam spot with very high precision to obtain a good estimate of $P_{Beam} \rightarrow M^2_{miss} = (p_{e^+} + p_{e^-} - p_\gamma)^2$.

Two MIMOSA 28 Ultimate Pixel Sensors [25] (size = $20.22 \times 22.71$ mm$^2$, thickness = 50 µm) are placed up- and down-stream of the diamond target in the cross before the large vacuum chamber (see Figure 6b) housing also the active target on the opposite extremity. The sensor active area spans 928 row $\times$ 960 column of pixels with a pitch of 20.7 µm.

The chip dissipates $\approx$ 150 mW/cm$^2$ and is normally operated at room temperature with air cooling. For PADME, it will be placed in a $10^{-4} \div 10^{-5}$ mbar vacuum: it is the first time this detector is being operated in vacuum, thus cooling is necessary via a Peltier cell attached to a brass finger (called "long support" in Figure 6b) placed in vacuum. Vacuum and temperature tests have been successfully performed and the detector is now operative.

The MIMOSA detector cannot be placed on the beam line during standard data taking because it would spoil the beam energy spread and divergence. Therefore, a downstream beam monitor is installed to provide a continuous run-time monitor of the beam: $2 \times 6$ Timepix3 sensors covering an area of the order of $10 \times 3$ cm$^2$ (see Figure 7).

The Timepix3 sensor [26] is a pixel sensors with an active area of 256 row $\times$ 256 column of pixels with a pitch of 55 µm each covering an area of about $14 \times 14$ mm$^2$. The sensor can record ToA (Time of Arrival) and ToT (Time over Threshold) simultaneously in each pixel with resolutions, respectively, of 1.56 ns and 10 bits. The sensor readout logic can stand a rate up to $80 \times 10^6$ hits/s thanks to eight differential data transfer links operating up to 640 Mbit/s each, summing an overall data rate up to 5.12 Gbit/s for a single sensor. Those performances allow without problems characterizing the bunches of 20,000 particles (positrons) with the length of about 200 ns used in PADME. The impinging coordinates and arrival time of each positron are measured. The integrated ToT measurement (considering the pixel pile-up) allows a precise estimation of the beam luminosity.

The number of positrons hitting the target over the entire data taking (nPOT) is a crucial parameter to measure cross sections and derive limits on couplings. Its measurement will be provided by the active target, which is able to measure the beam profile in both the the X and Y view at each bunch crossing.

By measuring both the number of tracks reaching the Timepix3 and the beam charge in the target, under the same beam conditions, we can obtain an absolute calibration of the measurement of the number of positrons hitting the target at each bunch.

In addition, Timepix3, providing imaging of the beam after the magnetic field deflection, allows estimating the beam divergence, spot size and time structure.

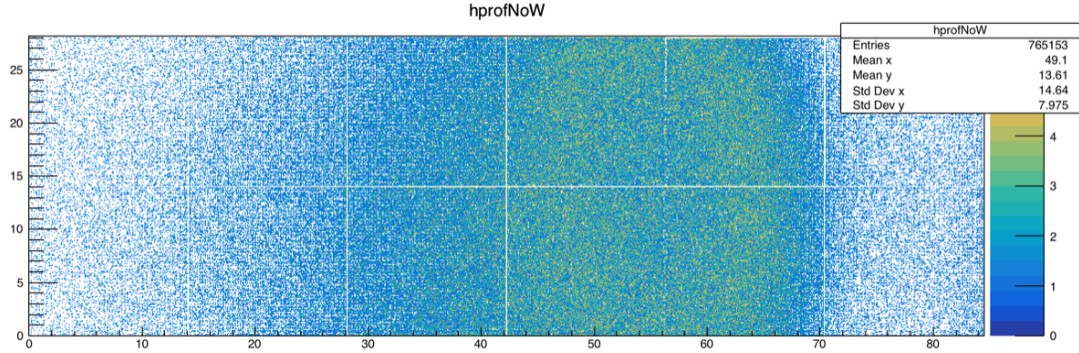

**Figure 7.** Timepix3 array showing the beam profile in the Data Acquisition.

## 5. Conclusions

The PADME experiment will search for an invisible "Dark Photon" (DP) at the the new dedicated BTF experimental hall built at the LINAC of the INFN Laboratori Nazionali di Frascati (LNF). The DP is predicted in a new class of physics models with a "hidden" sector.

Beam tests were made at LNF BTF in 2016 and 2017 for all the main detector components and all behaved as expected from design.

At the end of September 2018, the construction and commissioning phase was completed and data taking started at the beginning of October 2018. Three months of stable data taking in 2018 are expected and an extension of the run in 2019 might take place.

The collaboration submitted a proposal to set up a modified version of PADME at the Cornell University Synchrotron (Ithaca, NY, USA) running from 2020 to 2021 with a 6 GeV high intensity almost continuous beam, thus improving both the mass range and the sensitivity for the DP search.

Results found for the DP will apply also to other hypothetical light particles such as Axion-Like Particles, Dark Higgs, new proto-phobic vector boson, etc.

**Author Contributions:** P. Albicocco, F. Bossi, B. Buonomo, R. De Sangro, D. Domenici, G. Finocchiaro, L.G. Foggetta, A. Ghigo, P. Gianotti, G. Piperno, I. Sarra, B. Sciascia, T. Spadaro, E. Spiriti, E. Vilucchi (*INFN Laboratori Nazionali di Frascati*), A.P. Caricato, F. Gontad, M. Martino, I. Oceano, F. Oliva, S.Spagnolo (*INFN Sezione di Lecce and Dip. di Matematica e Fisica, Università del Salento*), C. Cesarotti, A. Frankenthal, J. Alexander (*Department of Physics, Cornell University*), G. Chiodini (*INFN Sezione di Lecce*), F. Ferrarotto, E. Leonardi, F. Safai Tehrani, P. Valente (*INFN Sezione di Roma*), S. Fiore (*INFN Sezione di Roma and ENEA*), G. Georgiev, V. Kozhuharov (*University of Sofia St. Kl. Ohridski and INFN Laboratori Nazionali di Frascati*), B. Liberti, C. Taruggi (*INFN Laboratori Nazionali di Frascati and Universitá degli Studi di Roma Tor Vergata*), G. C. Organtini, M. Raggi (*INFN Sezione di Roma and Dip. di Fisica, Sapienza Universitá di Roma*), L. Tsankov (*University of Sofia St. Kl. Ohridski*).

**Funding:** This work was partly supported by the Italian Ministry of Foreign Affairs and International Cooperation (MAECI), under the grant PRG00226 (CUP I86D16000060005).

**Acknowledgments:** We acknowledge the LNF tecnical staff for their help in the realization, construction and commissioning of the detectors, and the LINAC and BTF staff for their invaluable support in the tests of the detectors and in running the LINAC for the experiment. The PADME collaboration wish to thank particularly the L3 Collaboration and S.C.C. Ting for providing us the BGO crystals from the dismantled L3 experiment.

**Conflicts of Interest:** The authors declare no conflict of interest.

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
