# Peer review of "The Investigation on the Dark Sector at the PADME Experiment"

_universe, doi:10.3390/universe5020059_

Round 1

Reviewer 1 Report

Very interesting presentation and update status of the PADME experiment which aims to shed light on the "particle" nature of dark matter. Physical motivation is clearly discussed and the experimental setup is presented in great detail. Any reader can have a reasonable idea about the PADME experiment and its scientific targets. I feel very confortable to recommend publication of this manuscript in UNIVERSE.

Author Response

Sorry for the delay but I was asked to review the manuscript on 24/12 just before my planned Christmas Vacations.

Thanks for your appreciation of the presentation.

Reviewer 2 Report

Referee report to the manuscript number: universe-407736

Authors:  Fabio Ferrarotto (on behalf of the PADME Collaboration)

Title: The Investigation on the Dark Sector at the PADME Experiment 

The paper described the physics motivation, experiment design, and experiment sensitivity for the PADME experiment, in which the experiment is designed to detect the dark photon. 

Before considering for publication, the author may need to clarify the current status about the dark-photon on muon g-2 and on 17 MeV measured in excited Beryllium. For instance, the parameter space of dark photon used to explain muon g-2 has been excluded by BarBar collaboration, see 1702.03327 [hep-ex]. In addition, NA62 collaboration recently reported the result for searching for the new boson of 17 MeV. Due to no evidence, a strict limit on the parameter is given in 1803.07748 [hep-ex]. 

If the parameter space for explaining muon g-2 in dark-photon model is excluded, should PADME use it as one of physics cases in the PADME proposal? Will the PADME constraint on the parameter space for a new boson of 17 MeV be stricter than or compatible with NA62 data? The positive and negative possibility should be stated clearly. 

Hence, the author should properly make some comments about the questions in the paper. 

Author Response

Sorry for the delay, but I was asked to review the manuscript on 24/12 just before my planned Christmas Vacations. This week we had also the Padme Collaboration meeting , so I profited to ask also my colleagues contributions for the answers to your questions.

Thanks for the appreciation of the paper. I will try to answer clearly to your observations.

- Point 1 : agreed : we should say something about the Babar measurement.

We can insert just after line 55 the following :

"As visible in Fig. 2 a) in 2017 Babar has put a stringent  limit on the dark photon production  phase space excluding the simplest models for the (g-2)_{$\mu$} contribution"

and citing the article : Phys. Rev. Lett 119,131804

The citation of g-2 has been one of the starting point for the proposal of the experiment, and besides

the BaBar limit only exludes the simplest productions model so far, so maybe we should clarify in the text that this was one of the original physics cases proposed for the experiment, but not exclude it yet.

- Point 2 : for the 16 MeV meson (By the way, the article you cited is from Na64, not Na62!)  

we can add after line 67 :

"Na64 experiment has recently put a limit on the production of this possible new meson, but still allows a fair window in the production phase space where Padme is able to llok for the production of such meson. Padme has the furtjer advantage, respect to a "beam-dump"-like experiment of being able to tag the associated decaying charged particles to measure the meson invariant mass."

and citing the article : arXiv 1803.07748

Our sensitivity for production of this meson at rest is at least as good (epsilon>10^-4) as the one for the dark photon, but since the cross section for production near threshold increases , I am not sure how much better can we be : it's difficult to judge without a complete MC for this process , which we

don't yet have.

I hope this may clarify our position on the subject. I would try not to go into too many details on the subject since this is not the main topic of the talk.

Reviewer 3 Report

The paper is a review on the "state-of-art" and the features of the PADME experiment; it summarizes the goal, the experimental set-up and the roadmap of this experiment.

This paper is worthy to be published in the Proceedings of the Conference, providing some minor corrections 

- line: 171 "engular" should be changed in "angular"

- Fig. 5b, to improve the readability can be helpful either to add the elements' names or to add in the caption what element correspond to a given color

- Fig. 5 and 7 should appear slightly earlier

- lines 252-254 About the sentence "This measurement however is affected by a systematic uncertainty arising from the unknown local charge ...",
  a more detailed comment is required. How large is ?

Author Response

Sorry for the delay, but I was asked to review the manuscript on 24/12 just before my planned Christmas Vacations. This week we had also the Padme Collaboration meeting , so I profited to ask also my colleagues contributions for the answers to ypur questions.

- point 1 : OK. sorry , my fault for not correcting before submission, but this escaped me.

- point 2 : OK: you are right : I will add in the caption the details you asked for.

- point 3 : I tried it already , but I ended up in Latex having 2 double figures in a page, that he tends to make as a page with basicly only figures. It seems to me that, despite the fact that fig 5 and 7 end up a little later in the text, intersperse figures and text improves the readability anyhow.

- point 4 : this was one of the points in our tecnical design report.

 I asked the original author about this and he agrees that we may drop the sentence altogether.

Just to clarify : when we have intercalibrated the target with timepix3 during our commissioning and during the physics run we have seen a global estimated 5 to 10% intercalibration effect (depending on the beam intensity)  which includes all factors, thus telling us this factor is not really a limiting one, as was feared at the time we have designed the apparatus.

Round 2

Reviewer 2 Report

The author has clarified my previous questions. Now, the paper is suitable for publication in Universe.